# Restenosis after Coronary Stent Implantation: Cellular Mechanisms and Potential of Endothelial Progenitor Cells (A Short Guide for the Interventional Cardiologist)

**DOI:** 10.3390/cells11132094

**Published:** 2022-06-30

**Authors:** Tommaso Gori

**Affiliations:** German Center for Cardiac and Vascular Research (DZHK) Standort Rhein-Main, Department of Cardiology, University Medical Center Mainz, 55131 Mainz, Germany; tommaso.gori@unimedizin-mainz.de; Tel.: +49-6131-172829

**Keywords:** coronary arteries, inflammation, stent, stent thrombosis, stent restenosis

## Abstract

Coronary stents are among the most common therapies worldwide. Despite significant improvements in the biocompatibility of these devices throughout the last decades, they are prone, in as many as 10–20% of cases, to short- or long-term failure. In-stent restenosis is a multifactorial process with a complex and incompletely understood pathophysiology in which inflammatory reactions are of central importance. This review provides a short overview for the clinician on the cellular types responsible for restenosis with a focus on the role of endothelial progenitor cells. The mechanisms of restenosis are described, along with the cell-based attempts made to prevent it. While the focus of this review is principally clinical, experimental evidence provides some insight into the potential implications for prevention and therapy of coronary stent restenosis.

## 1. Introduction—Endothelial Progenitor Cells and Stent Failure

Implantation of coronary artery stents is the most commonly used strategy for the treatment of coronary artery disease [1]. Even though their application leads to improved quality of life and survival for millions of patients each year, these therapies are also associated with failures that limit their long-term efficacy. After percutaneous interventions (PCI), the most common cause of long-term failure is in-stent restenosis (ISR). ISR occurs with a frequency that was reported to be as high as 25% to 50% in bare metal stents (now discouraged in the current guidelines); although its rate has become significantly lower with the introduction of drug eluting stents (DES) [2], DES failure remains a problem that affects, depending on a number of factors, up to 20% of the devices implanted [1]. Beyond the implications for the treatment of these events (the majority of patients requiring further myocardial revascularization procedures), percutaneous treatment of ISR is also associated with a much higher rate of re-failure than treatment of native vessels. Procedural issues such as malaposition or incomplete stent expansion disrupt laminar flow and induced neointima proliferation [3]. In addition, the mechanical trauma caused by the stent implantation, hypersensitivity foreign-body reactions, the thrombogenicity of the stent surfaces which leads to adhesion of platelets and release of chemotactic and inflammatory substances, and, over more prolonged follow-up, progression of atherosclerosis in the stent neointima are all accepted mechanisms that concur with ISR [4,5,6]. Progenitor cells (PCs) are a heterogeneous group of blood cells that are believed to be involved in repair processes [7,8,9]. Their role in restenosis and as potential instruments in improving the outcomes of PCI remains unexploited, but interesting attempts have been made. This review will summarize the cellular mechanisms involved in the pathophysiology of stent failure and the potential of PCs to prevent it.

## 2. Cells Involved in the Pathophysiology of In-Stent Restenosis

Bare metal stent restenosis has affected as many as 50% of the devices implanted. Progressive growth of a fibrotic extracellular matrix resulting in a slowly progressing negative remodeling of the lumen was the typical feature of this complication [2]. In the DES era, the biochemical and cellular mechanisms involved in ISR appear to depend on hypersensitivity and foreign-body reactions, inhibition of healing mechanisms, as well as on proliferative responses triggered by the vascular damage during stent implantation [10,11]. Vascular smooth muscle cell (SMC) proliferation and migration, the synthesis of an extracellular matrix, and the migration of bone-marrow PCs appear to drive its pathophysiology [10,11]. A combination of both local and systemic responses to vessel damage determines whether vascular healing will result in the formation of a protective, smooth and thin, neointima or a restenosis.

Locally, the vascular injury caused by stenting triggers a complex cascade of events that include endothelial denudation, exposure of prothrombotic intima and subsequent inflammation, release of growth factors and cytokines, platelet activation, and SMC proliferation and migration. The result of these processes may be healing or pathological processes such as excessive neointimal hyperplasia (timeframe 6–12 months typically) or neoatherogenesis (timeframe > 12 months after PCI), which ultimately cause restenosis. The endothelial activation that follows injury causes degranulation and aggregation of platelets, which are probably the first cellular type involved in the reactions to stents [12,13]. These mechanisms have been summarized recently in a review by Neubauer and Zieger [14]; in short, endothelial injury leads to an increased local bioavailability of ATP, ADP and AMP and a decreased activity of ectonucleoside triphosphate diphosphohydrolase-1 (E-NTPDase1/CD39), a membrane-bound enzyme that converts ATP and ADP into adenosine. Soluble ADP triggers the recruitment and aggregation of platelets during the formation of a hemostatic plug. In addition, Antithrombin III (which normally binds to heparan sulfates of proteoglycans in the endothelial glycocalyx) and thrombomodulin, a membrane-bound thrombin receptor and inhibitor of the thrombin pathway are depleted following endothelial injury, removing two potent inhibitors of platelet activation. All these processes are important pharmacological targets.

In an observational study of 161 patients with recurrent angina after coronary stenting, platelet distribution width, an easily accessible marker of platelet activation, was an independent predictor of stent restenosis (Odd’s ratio 1.2 [1.02–1.4], *p* = 0.025) [15]. Following arterial injury, platelets rapidly adhere and release thromboxane A2; the glycoprotein (GP) IIb/IIIa complex binds to fibrinogen, leading to platelet aggregation and activation [16]. In addition, activated platelets release multiple factors including platelet-derived growth factor (PDGF), which exerts a mitogen and chemotactic effect on SMCs and contributes to oxidative stress, one of the major determinants of the switch of vascular SMCs from a contractile to a synthetic phenotype [17]. Changes in multiple parameters reflective of oxidative stress including malondialdehyde, superoxide dismutase and paraoxonase-1 in patients with ISR confirm the importance of this process [18]. In a randomized trial from our group, potent platelet inhibition with prasugrel (as compared to ticagrelor or clopidogrel) was associated with reduced markers of oxidative stress and it prevented stent-induced systemic endothelial and microvascular dysfunction [19] (which in turn are associated with long-term in-stent restenosis [20]). Histamine released from platelets and mast cells may also cause intimal hyperplasia, as the levels of this mediator were shown to increase by 20 to 90 times in a porcine model of endothelial injury, while in vitro histamine potentiated the PDGF-stimulated proliferation of cultured SMCs [21]. Platelet-derived interleukin-1 increases production of proinflammatory interleukin-6 and interleukin-8, respectively, stimulating SMC migration by actin polymerization and tyrosine phosphorylation of focal adhesion-associated cytoskeletal protein and the proliferation of SMCs [22]. In this context, thrombin also has potent mitogen effects, and the immunosuppressive effect of the drug eluted might slow tissue repair reaction and fibrin removal. The prolonged activation of the thrombotic cascade concurs with the activation of inflammatory cells (monocytes, T cells, neutrophils), secretion of chemoattractant molecules (MCP-1 or monocyte chemoattractant protein-1, Interleukin (IL)-8) by endothelial and SMCs and the production of growth factors (PDGF, platelet-derived growth factor, bFGF or fibroblast growth factor, TGF or transforming growth factor-beta, IGF, insulin-like growth factor, and VEGF, vascular endothelial growth factor [23]). Platelet-derived extracellular vesicles stimulate SMCs to produce interleukin 6 and express α_IIb_β_3_ and P-selectin, which favor their interaction with monocytes [24]. In addition, platelet-derived microvesicles stimulate endothelial (vWF+ and CD34+) PC proliferation by delivering transforming growth factor (TGF)-β1 in a rat model of arterial injury [25]. Finally, the role of plasma small extracellular vesicles containing miRNA-501-5p of endothelial origin has been demonstrated, as these promote SMC synthesis and are increased in patients with restenosis [26,27].

In the vessel wall, inflammatory responses to the foreign body and the antiproliferative effect of the drug eluted by the stent lead to delayed reendothelialization. Exposure of adhesion molecules (P-selectin, intercellular adhesion molecule-1) stimulates the recruitment of monocytes and the subsequent secretion of inflammatory cytokines (interleukins 6 and 8), resulting in the influx of neutrophils, monocytes and macrophages into the subendothelial space [22]. In line with this, elevated levels of monocytes (odds ratio 1.44, 95% CI: 1.23–1.68, *p* < 0.001) and eosinophils (odds ratio: 1.22, 95% CI: 1.09–1.36, *p* = 0.001) at three months after PCI are predictors of late in-stent restenosis after drug-eluting stent implantation [28]. The mechanisms that follow have been demonstrated in animal models of ISR: The release of cytokines and growth factors trigger the migration and proliferation of SMCs. Indeed, macrophages’ matrix metalloproteinase (MMP) 8 protein promotes SMC differentiation and matrix (neointima) production from adventitial stem/progenitor cells by modulating transforming growth factor-β activity and ADAM10/Notch1 signaling [29]. On the endothelial surface, downregulation of thrombomodulin and upregulation of tissue factor, fibrin, proteoglycan and collagen deposition occur. Von Willebrand factor (vWF), collagen and fibrinogen exposed on the endothelial cells surface and in the injured sites attract platelets that degranulate and secrete mediators, express adhesive proteins (fibrinogen, fibronectin) and surface glycoprotein receptors [30,31]. Following endothelial activation by mechanical trauma, reactive oxygen species are also generated, which causes vasoconstriction and activation of a prothrombotic phenotype with further increased synthesis of various cytokines [32]. Finally, in addition, the drugs delivered by the stents, such as paclitaxel and sirolimus, have been shown to induce tissue factor and PAI-1 expression and inhibit tPA and induce apoptosis, and to stimulate oxidative stress, which further delays endothelial restoration and function [33]. Although their role in restenosis remains unclear, mast cells release chymase, which lead to the production of angiotensin II and tumor growth factor-β, resulting in fibroblast proliferation and neointimal formation [34,35]. Inhibitors of chymase expression indeed inhibit neointimal formation in animal models [36].

## 3. The Role of Progenitor Cells

Importantly, while surely causing proliferation, all these inflammatory mechanisms might also have additional effects mediated by the recruitment of PCs (PCs). The processes mediated by the recruitment of these cells are also complex, and may work both in the direction of favoring reparatory processes but also in promoting restenosis.

The concept that stent implantation might result in the mobilization of PCs and that these cells might be involved in reparative processes but also in hyperplasia was proposed in the late 1990s and early 2000s [37,38,39]. In their original murine model of carotid injury, Werner et al. demonstrated that administration of rosuvastatin enhanced the colonization of bone marrow-derived endothelial cells expressing von Willebrand factor and cadherin in the injured vessel wall reducing neointimal formation [40]. Similarly, direct intravenous transfusion of spleen-derived endothelial PCs double positive for DiI-Ac-LDL and lectin was associated with homing of the transfused cells to the injury site, and lectin binding confirmed their endothelial phenotype. Reduced neointima formation was also observed in transfused animals [41]. Of note, the recruitment of PCs to sites of injury is also mediated and supported by platelets; in animal models, platelet-derived mediators such as the stromal cell-derived factor-1 or the platelet-derived factor AB promote the adhesion and differentiation of human CD34+ cells (a rather undifferentiated form of PC which has the potential to differentiate into endothelial cells but also into both a myeloid and a lymphoid lineage) to the endothelium, favoring their homing to sites of injury [42]. In addition, in vitro, platelet aggregates facilitate tethering and rolling of CD34+ cells via P-selectin binding; fibrin-containing thrombi attract CD34+ cells to the site of injury, promoting their differentiation toward a mature endothelial cell phenotype [43].

Importantly, other studies have provided evidence against the role of PCs in arterial repair. In the mouse model of Hagensen et al. and Tsuzuki, homing of circulating cells was not observed and the principal mechanism of repair was migration of flanking endothelial cells [44,45]; in the study by Tanaka, the origin of reparatory cells was dependent on the model of injury used and not the same for all models [46].

Significant controversy also exists regarding the origin of PCs, their role and function once they migrate into the vessel. With regard to the origin, while the bone marrow may produce cells that support angiogenesis in vessel walls, it does not appear to supply cellular populations that mature into endothelial cells [45,47], and it has been proposed that PCs may arise directly from a subadventitial “vasculogenic zone” [48]. As briefly described above, adventitial stem/progenitor cells have been attributed an important role in neointima proliferation following arterial injury [49]; however, their role as a target of therapy or prevention remains unexplored. With regard to the type of cells designated as PCs, the CD (“cluster of differentiation”, also known as “cluster of designation” or “classification determinant”) protocol is used for the identification of the different cell types based on their surface molecules. Applied to the classification of endothelial PCs, endothelial PCs were initially identified as CD34+ cells (as opposed to “shedded” mature endothelial cells, which are CD34− (but KDR+, CD31+, VEGFR2+)). These CD34+ early-outgrowth cells, also called “endothelial cells colony forming units” [50], originally identified as a potential mechanism of endothelial colonization and regeneration, are indeed recruited after vascular injury [51,52], but their proliferative potential and capacity to form mature endothelial cells is limited. While these cells still support vascular repair indirectly via phagocytic and paracrine secretory activities, more differentiated PCs, the late-outgrowth endothelial PCs [47], are capable of maturing into functioning endothelium. These cells have a higher proliferative potential and express CD31 and KDR, while disagreement exists regarding whether cells identified by their CD133 [53] may mature into endothelial cells or not.

Beyond endothelialization and neointima production, PCs, therefore, also have regulatory functions as they release paracrine mediators, including the vascular endothelial growth factor or the hepatocyte growth factor that are involved in the vascular repair processes by activating both resident endothelial cells and fibroblasts to produce an extracellular matrix [54]. Although proliferation of SMCs and production of the intercellular matrix are the main drivers of neointima formation and restenosis following percutaneous interventions, other types of PCs have a regulatory effect on these cells. For instance, endogenous c-kit^+^ stem/PCs (which itself do not differentiate into endothelial or SMCs) evolve into monocytes/macrophages and granulocytes, which modulate vascular immuno-inflammatory responses to endothelial injury [55]. Similarly, exposure of SMCs to PC mediators or microparticles causes them to increase their production of an extracellular matrix [56] while direct cell–cell contacts between PCs and SMCs inhibit these changes and actually protect from the induction of a synthetic phenotype by cholesterol loading [56]. Thus, the role of PCs in the pathophysiology of coronary restenosis after PCI is complex, as PCs, in different contexts, may have either beneficial or detrimental effects (Figure 1).

## 4. Progenitor Cells Counts and Outcome after Stenting

A limited number of studies have investigated the relationship between PC counts and restenosis or cardiovascular outcomes after stenting. Although it is well accepted that PCs are involved in vessel repair and may play a role in post-PCI processes, discordant results regarding the quantitative relationship between PC numbers and extent of restenosis have been reported. Of note, most of these studies were too small to provide definitive evidence. In addition, a number of factors may influence endothelial PC count and, therefore, act as confounder. These include, for instance, age and gender, diabetes, family history, hypertension and hypercholesterolemia, all of which are associated with a quantitative reduction and functional inhibition of PCs (both mature, i.e., CD34+/CD133+/KDR+, and less mature, i.e., CD34+) [55,57,58,59,60]. It is believed that a dysfunction of PCs may at least partially explain the association between patient characteristics such as age, diabetes mellitus, hyperlipidemia and renal impairment and the risk of restenosis [61]. On the other side, the administration of nitric oxide donors such as organic nitrates and/or physical exercise may actually increase the counts of CD133+, CD34+ and vascular endothelial growth factor receptor positive cells, as well as their proliferation activity, adhesion and migration ability in patients with coronary artery disease [62]. Finally, the impact of various other pharmacological interventions on the number and function of PCs has also been shown, including statins, angiotensin converting enzyme inhibitors, angiotensin blockers, beta-blockers and clopidogrel [63,64]. Risk factors also interact with these responses. In diabetic patients, Lee et al. demonstrated no increase in PC levels after elective PCI, an observation which was later confirmed by Fadini et al. [65,66].

With bare-metal stents, post-PCI increases in CD43+, or pre-PCI higher levels of CD133+, and CD14+ PCs have been proposed to identify patients at higher risk of developing restenosis in the reports by Inoue et al. and Pelliccia et al. (*n* = 40 and *n* = 155) [67,68]. Interestingly, in the Inoue paper, mononuclear PCs differentiating into both endothelial cells and alpha-smooth muscle actin-positive smooth muscle-like cells were observed. While the former may result in stent healing, the latter may contribute to restenosis. Implantation of sirolimus-eluting stents suppressed both types of differentiation. Notably, the same authors also reported differences in the counts of CD34+/CD133+/CD45^null^ and CD34+/KDR+ cells based on the stent type (with a larger response for a newer-generation ultra-thin strut DES as compared to a second-generation DES) group. Neointimal thickness and neointimal coverage rate were proportionally greater in the former [69]. In the paper by Pelliccia, 30 of the 155 patients showed in-stent restenosis at 8 months. The absolute counts of CD34+/KDR+/CD45− cells (i.e., progenitors of endothelial lineage) and of CD133+/KDR+/CD45− cells (i.e., progenitors of endothelial cells at an earlier stage) were significantly higher in these patients. In the study by Schober et al. (*n* = 17), CD34 + PC counts increased after PCI in patients who later developed restenosis, while they actually decreased in patients with good outcomes [70]. In the paper by DeMaria et al., patients (*n* = 20) undergoing elective PCI were randomized to bare metal stent alone or with drug coated balloon; neointimal hyperplasia and percentage were linearly correlated with CD34+ CD45dimKDR+ EPC levels at the time of implantation, while the percentage of uncovered struts was negatively associated, which confirms the dual role for these cells in determining stent endothelialisation but also neointimal growth. Other investigations reported a negative, or nonexistent association of neointimal hyperplasia with CD34+ PC counts [71].

With drug-eluting stents, Sakuma et al. observed an increase in circulating CD34+ CD133+ CD45 low cells and serum levels of biomarkers relevant to stem cell mobilization in patients treated with bare metal stents but not in those treated with DESs. In line with this, the percentage of uncovered struts was higher in the DES group compared with the BMS group, and PC numbers correlated with neointimal area. In vitro, sirolimus, zotarolimus and everolimus dose-dependently inhibited the differentiation of mononuclear cells into endothelial-like cells [72]. In the study by Montenegro et al. no significant association between PC counts (defined as CD45− CD34+ CD31+ CD133/2+ CD309+ cells) and the incidence of restenosis was evident, even though there was a trend towards higher PC counts before and after PCI in patients who would later develop restenosis. In this study, a decrease in PC counts was observed in two-thirds of the patients [73]. The size of the study (*n* = 37) might, however, limit the capacity of this study to detect differences. Finally, in a meta-analysis including nine studies, lower baseline CD34+ PC counts were associated with a significantly greater occurrence of in-stent restenosis (HR 1.33; 95% CI 0.97–1.82, *p* = 0.045) [74]. Of note, procedural characteristics (beyond the type of stent implanted) might also act as confounders. In a recent study in patients undergoing DES implantation, Jimenez-Quevedo et al. found that the systemic response (increase in the circulating CD133+/KDR+/CD45 low cells at 1 week after stenting) was proportional to the degree of vessel wall injury as manifested by intravascular imaging criteria (number of dissections, the number of quadrants with dissections; the length of edge dissections; and the number of dissections at the borders); in addition, these parameters of vessel wall injury were shown to be prognostic determinants of in-stent neointimal growth at 9 months [46]. In sum, PCI and arterial injury lead to mobilization of PCs, but the influence of risk factors (which are associated with reduced PC counts but also with increased risk of restenosis) and the type of stents implanted (modern ones being less traumatic and more biocompatible) complicate this relationship.

## 5. Stents Capturing Endothelial-Progenitor Cells

While the discussion of new stent platforms with more biocompatible (or no) polymer, thinner struts, streamlined design, goes beyond the scope of this paper, cell-based interventions have also been proposed to facilitate stent healing.

With the advent of DESs, the introduction of cytostatic or cytotoxic drugs that produce the above-described long-lasting inflammatory responses might be associated with long-term restenosis. This concept led to the development of bioengineered platforms featuring an antibody able to capture PCs on their luminal surface with the goal that these cells might mature into endothelial cells and favor healing of the stent surface. Upon implantation, the increased rate of capture of circulating PCs would promote endothelialization, reducing the risk of restenosis and thrombosis (and potentially leading to a reduced need for dual antiplatelet therapy). In an effort to improve stent healing, stainless steel stents coated with monoclonal antibodies anti-CD133, CD34 and CD146, i.e., able to bind and retain endothelial PCs, were developed. The first device of this type was the Genous monoclonal anti-human CD34 coated stent. This EPC-capture bare metal stent was covered by a covalently coupled polysaccharide polymer with embedded CD34 antibodies. In vitro studies confirmed that CD34 antibodies stents promoted endothelial coverage. In a porcine model, endothelialization (as assessed by scanning electron microscopy and histology) at 14 and 28 days was increased in the groups which received anti-CD34 antibody coated stents and HC-anti-CD34 antibody coated stents combined with sirolimus-eluting stents as compared to a “first generation” stent eluting sirolimus alone (respectively, 97 ± 3%, 95 ± 4%, 74 ± 8%; *p* = 0.042). Neointima proliferation at 28 days was also reduced in the group receiving combined therapy or sirolimus alone, but it was larger in those who had received anti-CD34 antibodies without sirolimus (neointima, in μm: 131 ± 44 vs. 185 ± 87 vs. 276 ± 108; *p* = 0.022) [75]. The first trial conducted with this device, the “Healing-FIM trial”, demonstrated an increased adhesion of CD34+ cells on the surface of the device as compared to standard bare-metal stents, but no difference in the amount of neointima hyperplasia during follow-up [76]. Further developments of PC-capturing stents included a combination of the use with a paclitaxel-coated balloon or with a DES [77]. The Combo stent (OrbusNeich Medical, Fort Lauderdale, FL, USA) features adluminal CD34-EPC capturing technology with abluminal release of sirolimus-eluting to minimize neointima formation. Preclinical trials in porcine models showed inhibition of neointima proliferation as compared to first-generation, thick-strut sirolimus-eluting stents (Cypher, Cordis, New Brunswick, USA) [78]. While clinical studies confirmed the non-inferiority in terms of safety and efficacy as compared to first-generation DESs up to 5 years and in single-arm registries [79,80], later lines of evidence demonstrated (in line with the results of the above porcine model) an increased rate of restenosis as compared to modern DESs with thin struts, biocompatible (or no) polymer, and “modern–limus eluting drug“ [81,82,83,84]. In the Japan–USA Harmonized Assessment by Randomized, Multi-Center Study of OrbusNEich’s Combo StEnt (HARMONEE) study, despite noninferiority of the Combo stent to Xience (Abbott Vascular, Santa Clara, CA, USA), a trend towards more frequent target vessel failure at 12 months in the COMBO group was observed. Similar data were reported by the SORT OUT X trial, a randomized, multicenter, single-blind, trial in which 3146 patients were randomized to the antibody-coated stent or to the sirolimus-eluting Orsiro stent (Biotronik, Bülach, Switzerland) [83]. In addition, in this study, despite noninferiority in terms of death, cardiac death and myocardial infarction at 12 months, the Combo stent led to an increased risk of target lesion revascularization. In the above mentioned meta-analysis, the increase rate of target lesion revascularization following implantation of the COMBO stent was confirmed [74]. Beyond differences in the DES technologies (e.g., different strut thickness, 100 μm for Combo and 60–81 μm for Orsiro or Xience), at least three hypotheses might contribute to explain these findings: the stent scaffold itself; the use of CD34, a surface antigen that is not exclusive to PCs and therefore might result in nonspecific binding of other PCs, with the potential of actually promoting neointima hyperplasia instead of inhibiting it; the fact that PC are reduced and dysfunctional in patients with coronary artery disease, as suggested by the fact that patients with higher titers of circulating CD34+ KDR+ EPC had lower rates of ISR compared to patients with reduced levels of PCs. This latter possibility appears to suggest that a combined strategy of stimulating the release of PCs from the bone marrow (see below) and improving the capture of these cells at the level of the stent might provide the desired benefit.

With regard to the second issue, stents covered with CD34 antibodies might actually attract cells uncapable of forming a new endothelial layer, paradoxically resulting in competition and reduced binding of “true” endothelial PCs (CD31+ or VEGFR2+). In a porcine model, a cobalt chromium stent coated with a CD31-mimetic peptide showed complete endothelialization already at seven days postimplantation, with no activated platelets/leukocytes, and neointima development at 28 days was significantly reduced compared to bare metal stents and showed, unlike DES, no thrombosis [85]. Other animal studies have addressed the hypothesis that combining CD133 with CD34, or using CD133 instead of CD34 capturing stents, might improve the endothelialization-fostering capacities of the platform. Using this strategy appears to lead to prolonged time of cell adhesion and numerically larger PC capture, enhanced endothelialization and reduced in-stent restenosis rates [86,87], but a CD133 stent was also associated with increased neointima proliferation in a porcine model [88]. Further studies focused on CD146, which is supposed to be expressed by late PCs, showed a further improvement in cell capture efficiency as compared to CD133 stents [89]. This advantage translated into a decreased neointimal rate and stenosis area, an increased lumen area (all improved by 30–60%) as compared to a bare metal stent in a porcine model. Whether this will translate in improved patient outcomes, and the effectiveness of other technologies such as magnetic molecules, aptamers and specific peptides [58,90] or in vitro endothelialized stents [91], will need to be tested in the future. 

## 6. Administration of PCs

The dual role of PCs—on one side as reparative cells, on the other as promoters of proliferation—is confirmed by studies investigating the effect of exogenous PC administration in settings of vascular injury. Following carotid wire injury in athymic mice, injection of PCs resulted not only in reduced neointima formation but also in altered cellular composition of the neointima with augmented accumulation of SMCs [56]. In a similar rat model, transplanted PCs promoted reendothelialization and inhibited neointimal hyperplasia through production of paracrine cytokines, while differentiation of PCs into mature endothelial cells was not observed [92]. In vitro, mechanical injury of SMCs was associated with the release of CXCL12, a key regulator for the mobilization of stem and PCs into peripheral blood and their subsequent recruitment to sites of tissue damage or stress, resulting in enhanced recruitment of EPCs to SMCs [93]. The question therefore arises, how to separate the positive effects of PCs from the negative ones. For instance, in an animal model of wire-induced endothelial damage, injection of PCs pretreated with netrin-1 (an axon-guiding protein that stimulates angiogenesis and cardioprotection via induction of the release of nitric oxide) substantially attenuated neointimal formation. EPC proliferation, nitric oxide production and resistance to oxidative stress-induced apoptosis were increased in this model [94]. In rabbits, implantation of a PC-capturing stent expressing anti-CD34 and anti-VEGFR-2 antibodies followed by PC infusion decreased neointima proliferation, in-stent restenosis and thrombus formation but had no impact on neointimal hyperplasia inhibition [95].

To date, no study has tested the effect of PC infusion in patients undergoing coronary stenting. However, a number of studies in patients with myocardial infarction, peripheral artery disease, pulmonary hypertension and heart failure have been performed with inconsistent results (Table 1). In the most recently published study, the effect of CD34+ cells infusions were tested in patients with microvascular angina [96]. Coronary microvascular angina is typical of patients with evidence of ischemia but without obstructive coronary artery disease; for these patients no specific therapy exists. In this recent study, 20 patients with microvascular angina received autologous CD34+ cell therapy, which resulted in an increase in coronary flow reserve from 2.08 ± 0.32 at baseline to 2.68 ± 0.79 at 6 months after treatment (*p* < 0.005) and a corresponding reduction in angina frequency (*p* < 0.004). In the setting of myocardial infarction, recent meta-analyses summarized the results of multiple trials conducted using different types of PCs. In the work of Wollert et al., including 12 trials, intracoronary cell therapy was associated with no improvement in patient prognosis or changes in left ventricular function [97]. In the meta-analsys of Xu et al., early (within 5 days after myocardial infarction) reinfusion of PCs in patients with reduced ejection fraction was associated with an improvement in contractility [98]. In sum, while PC therapy appears to be safe, its efficacy is at least controversial and its effects in setting of PCI have not yet been tested.

## 7. Conclusions

Percutaneous coronary interventions cause mechanical injury and vascular inflammation. Additionally, the presence of a foreign body and the proinflammatory effects of the polymer and the drug eluted by the stent stimulate complex processes involving endothelial cells, SMCs, platelets and inflammatory cells as well as endothelial PCs. These cell types are involved in both repair phenomena but also neointimal proliferation and restenosis. To date, stent platforms covered with anti-(CD43+)-PC have not proven superior (and in several studies have shown inferiority) as compared to modern thin-strut biocompatible DESs. Platforms coated with other antibodies appear to provide more positive results, but clinical data are missing. The potential of other approaches (including PCs of non-endothelial lineage) also remains to be tested. While studies exist in the setting of myocardial infarction and heart failure, currently no data are available on the effect of autologous re-transfusion of (non-endothelial) PCs following PCI. While modern DES appear to be safe and effective, new paradigms of therapies, aimed at exploiting the regenerative potential of endothelial PCs, still find no application.

## Figures and Tables

**Figure 1 cells-11-02094-f001:**
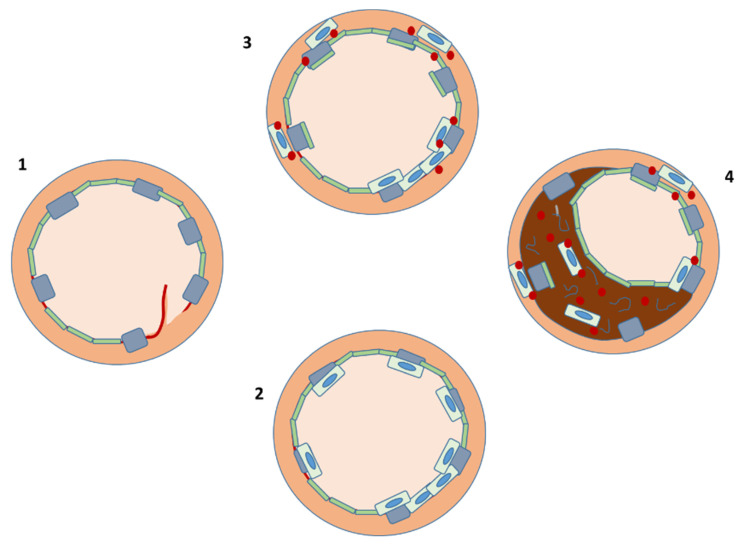
PCI results in arterial injury and exposure of a subendothelial proinflammatory environment (**1**). Circulating PCs are then attracted to the site of vascular injury and support repair by (**2**) transdifferentiating into endothelial cells or (**3**) paracrine production of chemokines (red), which stimulate native endothelial cell proliferation. The same processes may, however, (**4**) lead to SMCs proliferation and switch to a synthetic phenotype, causing restenosis.

**Table 1 cells-11-02094-t001:** Studies of PC therapy early after stenting for AMI.

Author, Year	Days from Reperfusion, (*n* of Patients)	Type of Trial	Type of PC	Outcome
**Schächinger 2004** [99]	5 (101)	Parallel, randomized	Autologous bone-marrow mononuclear cells	Improved LVEF
**Wollert, 2004** [100]	5 (39)	Randomized	Autologous bone-marrow mononuclear cells; <1% CD34+	Improved LVEF
**Lunde, 2006** [101]	6 (50)	Randomized	Autologous bone-marrow mononuclear cells; ~1% CD34+	No effect
**Tendera, 2009** [102]	7 (80)	Randomized	CD34+, CXCR4+ (endothelial)	No effect
**Roncalli, 2011** [103]	9 (52)	Randomized	Autologous bone-marrow mononuclear cells	SPECT evidence of improved viability
**Traverse, 2011** [104]	17 (58)	Randomized	Autologous bone-marrow mononuclear cells; ~2.5% CD34+, ~1% CD133+	No effect
**Makkar, 2012** [105]	~62 (17)	Parallel	CD105+	MRI evidence of improved viability
**Traverse, 2012** [106]	3 or 7 (79)	Randomized	CD34+ (1% CD34+, CD133+)	No effect
**Wöhrle, 2013** [107]	6 (29)	Randomized	CD34, CD45, CD133 and vascular endothelial growth factor-R2	No effect
**Sürder, 2013 and 2020** [108,109]	5 (51)	Randomized	CD34+, CD133+	No effect
**Peregud-Pogorzelska, 2020** [110]	1 (15)	Parallel, non-randomized	Autologous lineage-negative (LIN-) stem/PCs	Earlier decrease in troponin and BNP levels, smaller LV diameter at 12 months.
**Nicolau, 2018** [111]	Not reported (83)	Double-blind, randomized, multicenter study	Autologous bone-marrow mononuclear cells	No effect

BNP: brain natriuretic peptide. LV: left ventricle.

## Data Availability

Not applicable.

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
