# Peer review of "Restenosis after Coronary Stent Implantation: Cellular Mechanisms and Potential of Endothelial Progenitor Cells (A Short Guide for the Interventional Cardiologist)"

_cells, 2022, doi:10.3390/cells11132094_

Round 1

Reviewer 1 Report

In this review article, Tommaso Gori provides a brief overview about the cellular mechanisms involved in the vascular healing response after stent implantation and the formation of a neointima leading to restenosis, including endothelial progenitor cells. Findings in animal models are briefly presented as well as findings from clinical studies examining stents capturing EPCs or following PC administration.

The topic of restenosis formation is undoubtedly complex and has been summarized in many previous review articles. Nevertheless, the specific topic of this review, the presentation of data on the role of progenitor cells in this process, lacks focus (e.g. clinical evidence? experimental data? both?) and detailed information on specific contents (e.g. many statements lack references). Also, only endothelial progenitor cells are touched, whereas experimental evidence also points to smooth muscle and bone marrow progenitor cells, as well as other (e.g. perivascular progenitor cells), which are not mentioned in the manuscript.

In addition, I have the following specific comments:

1.     The majority of the cited literature or the data in Table 1 is from the early 2000’s and it is unclear to me what makes this particular topic timely? Of the 425 existing publications available in PubMed following a search with the terms “progenitor cells & restenosis”, approximately 25% (i.e. 98) are review articles. The added value and new information of this specific review article as compared to the 10 other ones from this year alone should be better worked out.

2.    Whereas the title and abstract of this review article mention “progenitor cells” in general, the text focusses entirely on progenitor cells of the endothelial cell lineage. Please adjust accordingly and also provide a rationale why only endothelial progenitor cells are presented here. Other potential vascular progenitor cell types that could play a role should be at least briefly mentioned.

3.  The title and the abstract should also contain information, whether the review focusses on clinical data or whether experimental evidence is also presented.

4.    Page 3, lines 104-106: there are also studies which did not find progenitor cells in vascular lesions. These also should be mentioned (e.g. PMID: 22012957, PMID: 19333644, PMID: 14500338). Also, citations should be provided for the first statement.

Minor:

·         Abstract: Please define or explain the term “hypersensitivity” (line 12). What exactly is meant by this?

·         Please check and correct for the absence of spaces before citations throughout the text, e.g. lines 26, 30 and so on.

·         Please check for typographical errors, e.g. lines 290, 292,

·         A half-sentence in lines 274-275 is highlighted in bold. What makes this statement so important compared to all others?

Author Response

[…]the presentation of data on the role of progenitor cells in this process, lacks focus (e.g. clinical evidence? experimental data? both?) and detailed information on specific contents (e.g. many statements lack references).

I apologize for this. In the revised version, I have taken care that statements are referenced, particularly in paragraph 3 and 4 (newly introduced). With regards to the focus of the paper (experimental vs clinical), the focus is definitely clinical and the goal is to provide a “simpler” paper that might be informative for practicing interventional cardiologists as now clearly stated in title and abstract. However, in several instances where clinical (human) data are not available and the only insight is provided by experimental evidence, this basic-science evidence is mentioned. In those cases, I have now pointed out throughout the paper that clinical evidence is missing.

  • Also, only endothelial progenitor cells are touched, whereas experimental evidence also points to smooth muscle and bone marrow progenitor cells, as well as other (e.g. perivascular progenitor cells), which are not mentioned in the manuscript.

You are absolutely right. This discussion has been expanded now (paragraphs 3 and 4).

In addition, I have the following specific comments:

  1. The majority of the cited literature or the data in Table 1 is from the early 2000’s and it is unclear to me what makes this particular topic timely? Of the 425 existing publications available in PubMed following a search with the terms “progenitor cells & restenosis”, approximately 25% (i.e. 98) are review articles. The added value and new information of this specific review article as compared to the 10 other ones from this year alone should be better worked out.

You are absolutely right, particularly because the clinical applications of these approaches remain unexploited. This was an invited review focusing on the potential of cell-based interventions to support vascular regeneration, and I felt that the setting of restenosis is particularly interesting. I tried to give the review a clinical cut, that may serve a short guide for interventionalists. As a member of this community, I always felt the need for more information on PCs (particularly EPCs), given the number of studies available, the availability on the market of a „EPC capturing stent“, and the lack of solid evidence supporting its use. In all sincerity, also my laboratory actually conducted a study with COMBO stents in chronically occluded vessels, and the results were not very satisfying (Reference Blessing et al). Had I read my own review before, probably I would have not performed the study.

  1. Whereas the title and abstract of this review article mention “progenitor cells” in general, the text focusses entirely on progenitor cells of the endothelial cell lineage. Please adjust accordingly and also provide a rationale why only endothelial progenitor cells are presented here. Other potential vascular progenitor cell types that could play a role should be at least briefly mentioned.

You are absolutely right. The manuscript indeed focuses on the endothelial progenitor cells because those are the only target of the interventions attempted (with little success) until now, at least in interventional cardiology. Since the focus of the review is clinical, I would indeed prefer to limit the discussion to these cells types. The title has been changed, and a short mention to the possible other types of PCs in vascular regeneration is made in Page 3.

  1. The title and the abstract should also contain information, whether the review focusses on clinical data or whether experimental evidence is also presented.

This has been done, thank you very much.

  1.   Page 3, lines 104-106: there are also studies which did not find progenitor cells in vascular lesions. These also should be mentioned (e.g. PMID: 22012957, PMID: 19333644, PMID: 14500338). Also, citations should be provided for the first statement.

Again absolutely right. The specific sentence in page 3 has been deleted (see also reviewer 3, as it was a repetition of the text above). The studies mentioned by you are very important and have been quoted in page 4.

Minor:

  • Abstract: Please define or explain the term “hypersensitivity” (line 12). What exactly is meant by this?

Thank you for this comment. By hypersensitivity I meant the processes described under paragraph 2. I do agree that this term, placed here, might be confusing. Therefore I removed if and left „inflammatory reactions“.

  • Please check and correct for the absence of spaces before citations throughout the text, e.g. lines 26, 30 and so on.

This has been done, thank you.

  • Please check for typographical errors, e.g. lines 290, 292,

This has been corrected, thank you.

  • A half-sentence in lines 274-275 is highlighted in bold. What makes this statement so important compared to all others?

I apologize, this was a typo.

Reviewer 2 Report

Here Dr. Gori considered the role of endothelial progenitor cells and the efficacy of EPC-capturing stents in preventing in-stent restenosis, a frequent and clinically significant complication of percutaneous coronary intervention. The review is generally of a high quality and can be accepted for publication upon the revisions.

Specific comments are provided below:

1. Page 2, lines 59-60: "The endothelial activation that follows injury causes degranulation and aggregation of platelets..."

Please describe the molecular mechanisms of endothelial-induced platelet degranulation/aggregation as this is a potentially targetable and critical point in endothelial dysfunction. This can be merged with a discussion below as this information is presumably provided there (lines 72-90), yet these 2 paragraphs now largely overlap.

2. Please also extend the discussion on "Platelet reactivity and size appear to be directly related to stent restenosis..."

Inherited and stochastic variation in platelet characteristics is an independent factor of their activity in pathophysiological scenarios (e.g., in-stent restenosis) and also should be better discussed.

3. According to the discussion provided at lines 66-70:
Please describe the paracrine effects of endothelial cells on vascular smooth muscle cells within the neointima, e.g. the effects of interleukin-6, interleukin-8, and MCP-1/CCL2 on a contractile-to-synthetic phenotypic switch.

4. Please describe the implied phenotype of endothelial progenitor cells, as they have very different subpopulations characterised by a distinct effects and immunophenotype (e.g. early and late progenitor cells). The conventional phenotype of endothelial progenitor cells is CD34+VEGFR2+CD133+CD45-. Did you mention this cell population as endothelial progenitor cells throughout the paper?

5. Section 3 "Progenitor cells counts and outcome after stenting": please provide a clear conclusion.

6. Lines 201-202: is 97±3% related to hyaluronan-chitosan-anti-CD34 antibody stents and 95±4% related to sirolimus-eluting stents? Please write it clearer. This not a significant difference in any case. What are the differences between "first-generation drug-eluting stents" and, for instance, sirolimus-eluting stents?

7. Line 214: again, what are "modern drug eluting stents". As soon as I understood from this sentence, they are opposed to sirolimus-eluting stents (which are also opposed to the first-generation drug-eluting stents at lines 201-202). Please describe the stent generations as clear as possible; there are many readers unfamiliar with their evolution.

8. Line 227: CD34 is a general stem cell marker and is not exclusive for endothelial progenitor cells; for this reason, the latter are additionally stained with VEGFR2 and CD133. Please emphasize that CD34, while being a common marker of progenitor cells, is not a specific marker of ENDOTHELIAL progenitor cells.

9. Upon the reading of the review, I am convinced that endothelial progenitor cells cannot be considered as a useful tool to prevent, hinder, or retard in-stent restenosis; neither pre-clinical nor clinical studies provide clear evidence for this hypothesis, and there are enough studies for all 3 outcomes (superior, non-inferior ot negative). Another strategy to clartify this hypothesis is the isolation of endothelial progenitor cells from the peripheral blood, their growth and differentiation in vitro and further intravenous re-administration in vivo. Yet, this is not described in the paper and I am not sure whether such attempts have been performed earlier.

10. Please demarcate "shedded", i.e. mature (CD31+VEGFR2+CD34-CD45-) circulating endothelial cells and endothelial progenitor cells (CD34+VEGFR2+CD133+CD45-). Another reason for inconclusive results of anti-CD34-coated stents is that they do not promote (instead, they presumably repel because of excessive adhesion of CD34+ cells) the adhesion of CD31+ cells (i.e., mature circulating endothelial cells shedded from the vascular endothelium). Possibly, anti-CD31 or anti-VEGFR2-covered stents could show better performance in pre-clinical and clinical studies.

11. Please also check references throughout the text as there are many sentences without the references (e.g. lines 114-116: "Similarly, exposure of smooth muscle cells to PC mediators or microparticles causes them to increase their production of extracellular matrix"). Provide reference after every sentence which is not your own discussion.

Author Response

[…] The review is generally of a high quality and can be accepted for publication upon the revisions.

Thank you very much for your positive evaluation.

  1. Page 2, lines 59-60: "The endothelial activation that follows injury causes degranulation and aggregation of platelets..."

Please describe the molecular mechanisms of endothelial-induced platelet degranulation/aggregation as this is a potentially targetable and critical point in endothelial dysfunction. This can be merged with a discussion below as this information is presumably provided there (lines 72-90), yet these 2 paragraphs now largely overlap.

I absolutely agree, a short discussion on the triggers for platelet activation has now been added (lines 67 and following).

Please also extend the discussion on "Platelet reactivity and size appear to be directly related to stent restenosis..." Inherited and stochastic variation in platelet characteristics is an independent factor of their activity in pathophysiological scenarios (e.g., in-stent restenosis) and also should be better discussed.

Thank you very much, this information has been better detailed (OR and size of the study) in page 2, last paragraph. Also, I added mention to our studies reporting a relationship between platelet function, endothelial function and microvascular reactivity.

According to the discussion provided at lines 66-70:
Please describe the paracrine effects of endothelial cells on vascular smooth muscle cells within the neointima, e.g. the effects of interleukin-6, interleukin-8, and MCP-1/CCL2 on a contractile-to-synthetic phenotypic switch.

Thank you very much, this section has been expanded (page 3, first paragraph). I have also made mention to reviews focusing on these themes.

Please describe the implied phenotype of endothelial progenitor cells, as they have very different subpopulations characterised by a distinct effects and immunophenotype (e.g. early and late progenitor cells). The conventional phenotype of endothelial progenitor cells is CD34+VEGFR2+CD133+CD45-. Did you mention this cell population as endothelial progenitor cells throughout the paper?

Thank you very much for this comment. I have now added a discussion of the different CD types and I have taken care to mention the cluster of cells identified as PC in each study.

Section 3 "Progenitor cells counts and outcome after stenting": please provide a clear conclusion.

Thank you very much, I have added a short conclusion: PCI causes PC mobilization, but the intereference of risk facotrs and type of stent complicates any conclusion regarding the clinical implications of this phenomenon in the routine.

  1. Lines 201-202: is 97±3% related to hyaluronan-chitosan-anti-CD34 antibody stents and 95±4% related to sirolimus-eluting stents? Please write it clearer. This not a significant difference in any case. What are the differences between "first-generation drug-eluting stents" and, for instance, sirolimus-eluting stents?

I apologize, this was not clearly formulated. I now presented the data in full (28 days). These data are also in line with those of the clinical studies that followed. The sirolimus eluting stent used in this study was indeed a cypher, ie a first-generation DES.

  1. Line 214: again, what are "modern drug eluting stents". As soon as I understood from this sentence, they are opposed to sirolimus-eluting stents (which are also opposed to the first-generation drug-eluting stents at lines 201-202). Please describe the stent generations as clear as possible; there are many readers unfamiliar with their evolution.

Thank you very much – I have expanded this sentence. Just to clarify: first-generation DES are the Cypher (Sirolimus-eluting) and Taxus (paclitaxel-eluting) stent, with thick struts and less biocompatible polymer. There is a large divergence with regards to the use of terms such as „second-generation“, „newer-generation“ etc stents, so I more clearly stated „modern stents, with thin struts, biocompatible (or no) polymer) and moder –limus eluting drug“.

  1. Line 227: CD34 is a general stem cell marker and is not exclusive for endothelial progenitor cells; for this reason, the latter are additionally stained with VEGFR2 and CD133. Please emphasize that CD34, while being a common marker of progenitor cells, is not a specific marker of ENDOTHELIAL progenitor cells.

This is very correct. I have emphasized this in page 3, third paragraph.

  1. Upon the reading of the review, I am convinced that endothelial progenitor cells cannot be considered as a useful tool to prevent, hinder, or retard in-stent restenosis; neither pre-clinical nor clinical studies provide clear evidence for this hypothesis, and there are enough studies for all 3 outcomes (superior, non-inferior ot negative). Another strategy to clartify this hypothesis is the isolation of endothelial progenitor cells from the peripheral blood, their growth and differentiation in vitro and further intravenous re-administration in vivo. Yet, this is not described in the paper and I am not sure whether such attempts have been performed earlier.

Absolutely right, to my knowledge there are no data on the effects of „EPC autotransfusion“. This has been added to the conclusions.

  1. Please demarcate "shedded", i.e. mature (CD31+VEGFR2+CD34-CD45-) circulating endothelial cells and endothelial progenitor cells (CD34+VEGFR2+CD133+CD45-). Another reason for inconclusive results of anti-CD34-coated stents is that they do not promote (instead, they presumably repel because of excessive adhesion of CD34+ cells) the adhesion of CD31+ cells (i.e., mature circulating endothelial cells shedded from the vascular endothelium). Possibly, anti-CD31 or anti-VEGFR2-covered stents could show better performance in pre-clinical and clinical studies.

Two very important points. The „shedded“ mature endothelial cells are mentioned in page 5, first paragraph. The possibility that CD34 antibodies might actually have a negative effect is also mentioned in page 8, last lines. The paper 10.1093/eurheartj/ehab027 is also mentioned here.

  1. Please also check references throughout the text as there are many sentences without the references (e.g. lines 114-116: "Similarly, exposure of smooth muscle cells to PC mediators or microparticles causes them to increase their production of extracellular matrix"). Provide reference after every sentence which is not your own discussion.

Thank you, I have now taken care that all statements are referenced. An additional 38 references have been added.

Author Response

[…] Without no doubt the review is a good and equilibrated summary of the current knowledge in this field, with a critical view on further directions. It is interesting, clear and well written.

Thank you very much for your positive evaluation.

  • In general there are some inconsistencies in abbreviation used to refer to endothelial progenitors. Mainly [PCs] is used, but, pe, in the definition of some populations (CD34+KDR+ EPC; lane 231), EPC is used. In addition is also referred to trials using CD34+ cells (lanes 273-275), they are totally equivalent. I would suggest to use [EPCs] for endothelial progenitors and [PCs] for less defined populations, as CD34+.

Thank you very much. I have now specifically added which cell populations were targeted in each study (ie CD34+), instead of calling them generally „EPCs“ or „PC“. Of course I still used the general definition „PC“ in generic statements such as „stents were developed to caputre endothelial PCs“, later specifying which type of antibodies were used in the different studies/platforms. I also avoided the abbreviation EPC to avoid confusion.

  • I am not convinced that the last paragraph (lanes 271-285) focused on the use of, main CD34+ cells but including other cell types, and evaluated for the treatment of AMI, (associated with table 2) is worth for the review.

This is a fair comment. I have changed the sentence focusing first on endothelial PC-capturing stents and then clarifying that thre is a potential also for other types of PCs (still not tested in the clinic).

  • Table. In case of maintainment of the table, it will interesting to include another column specifying the type of cell used. The most recent publication referred is from 2013. Is there no further clinical evaluation?

This has been done. Yes, a number of other publications have been added publications, including two from 2020.

  • Minor issues 1. When appropriate use platelets instead of thrombocytes.

This has been corrected, thank you

  • After the initial definition use always the corresponding abbreviature (SMC, DES, EPC).

Thank you, this has been done.

  1. Lanes 272-274. This paragraph is not clear on the nature of cell used (EPCs, PCs or both) “However, a number of studies in patients with myocardial infarction, peripheral artery disease, pulmonary hypertension and heart failure have been performed, with alternate results (Table 2).

You are right. I have added a column to the Table to detail the type of cells used.

  1. Table 2 is referred (lane 274); but there is no a Table1?

I apologize, a Table 1 was planned but never included.

  1. Probably this reference should be included Haude et al. The REMEDEE trial: 5-Year results on a novel combined sirolimus-eluting and endothelial progenitor cells capturing stent. Catheter Cardiovasc Interv . 2020;95(6):1076-1084. 6. Spelling:

Absolutely, thank you.

Lane 223. “rat” ---- rate?

Lane 273. “alternate” ---- I do not if this is the best qualification of referred results.    

Lane 290. “proinflammaotry”--- proinflammatory

Lane 292. “endothelial” ----- endothelial

Thank you very much, there errors have been corrected.

Round 2

Reviewer 1 Report

Thank you for carefully revising your manuscript taking into account my suggestions. I have no further comments.

Reviewer 2 Report

Dr. Gori has well addressed all my comments. The paper can be accepted for publication now.